# The Plug-in Approach for Average-Reward and Discounted MDPs: Optimal Sample Complexity Analysis

**Matthew Zurek**    MATTHEW.ZUREK@WISC.EDU  and  **Yudong Chen**    YUDONG.CHEN@WISC.EDU
*Department of Computer Sciences, University of Wisconsin-Madison*

**Editors:** Gautam Kamath and Po-Ling Loh

## Abstract

We study the sample complexity of the *plug-in approach* for learning $\varepsilon$-optimal policies in average-reward Markov decision processes (MDPs) with a generative model.[1] The average-reward plug-in approach estimates the parameters of the MDP model and then computes the optimal average-reward policy for the estimated model. This is arguably the most natural model-based approach for solving average-reward MDPs, yet its finite-sample properties have never been theoretically examined. Not only does our analysis fill a major gap in our understanding of a basic algorithm for this problem, but also we show that this algorithm is optimal in several settings without requiring strong assumptions about prior knowledge of the problem, thus addressing many limitations of previous approaches.

More specifically, the plug-in method, when combined with the stabilizing *anchoring* technique which has previously appeared in the average-reward reinforcement learning literature, can simultaneously achieve the optimal diameter- and mixing-based sample complexities. In particular, letting $D$ be the diameter of the MDP and $\tau_{\mathrm{unif}}$ the uniform mixing time, we show that this algorithm achieves the minimax-optimal sample complexities of

$$\widetilde{O}\left(SA\frac{D}{\varepsilon^2}\right) \quad \text{and} \quad \widetilde{O}\left(SA\frac{\tau_{\mathrm{unif}}}{\varepsilon^2}\right)$$

for learning an $\varepsilon$-optimal policy, *without needing to have prior knowledge of $D$ or $\tau_{\mathrm{unif}}$* and without needing to tailor the algorithm to the particular situations.

The above results are corollaries of our bias-span-based complexity bounds for weakly communicating MDPs, for example $\widetilde{O}\left(SA\frac{\|h^\star\|_{\mathrm{span}}+\|\widehat{h}^\star\|_{\mathrm{span}}+1}{\varepsilon^2}\right)$, where $\|h^\star\|_{\mathrm{span}}$ is the optimal bias span and $\left\|\widehat{h}^\star\right\|_{\mathrm{span}}$ is the (random) optimal bias span in a certain estimated MDP. We further show that the analysis behind this span-based bound (and several variants) is unimprovable, in the sense that the term $\left\|\widehat{h}^\star\right\|_{\mathrm{span}}$ cannot be removed in general for the performance of the plug-in method. In particular, this implies that the average-reward plug-in approach cannot generally match the minimax complexity of $\widetilde{O}\left(SA\frac{\|h^\star\|_{\mathrm{span}}+1}{\varepsilon^2}\right)$, which is only known to be attainable using the discounted plug-in approach with a carefully tuned discount factor requiring knowledge of $\|h^\star\|_{\mathrm{span}}$ (Zurek and Chen, 2025). We also show that by using a relatively smaller discount factor independent of $\|h^\star\|_{\mathrm{span}}$, it is possible to achieve a sample complexity of

$$\widetilde{O}\left(SA\frac{\|h^\star\|_{\mathrm{span}}^2+1}{\varepsilon^2}\right),$$

representing the first result with a per-state-action-pair complexity bounded in terms of only $\|h^\star\|_{\mathrm{span}}$ and *without requiring prior knowledge of $\|h^\star\|_{\mathrm{span}}$*.

---

1. Extended abstract. Full version appears as Zurek and Chen (2024).

While the average-reward plug-in approach can be seen as a large-discount-factor limit of the discounted plug-in approach, previous discounted analyses are incapable of being adapted to this problem. We address this limitation by developing several novel techniques for analyzing the error of long-horizon problems, which may be broadly useful. In particular, these techniques lead to improved results for the discounted plug-in approach, including removing effective-horizon-related sample size restrictions of previous results which achieve quadratic dependence on the effective horizon for the fixed MDP setting. Our techniques also enable the first optimal complexity bounds for the full range of sample sizes without the need for reward perturbation.

**Keywords:** Reinforcement learning, average-reward Markov decision processes, sample complexity, plug-in approach, without prior knowledge

## Acknowledgments

Y. Chen and M. Zurek acknowledge support from National Science Foundation grants CCF-2233152 and DMS-2023239.

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
