# OpenReview forum: "The Plug-in Approach for Average-Reward and Discounted MDPs: Optimal Sample Complexity Analysis"
_algorithmiclearningtheory.org/ALT/2025/Conference — ALT 2025_

### Official Review · Reviewer_DpK8 · 2024-10-24
**This work studies Average-Reward MDPs and achieves a near-optimal regret guarantee without prior knowledge of the diameter.**

**Rating:** 6
**Confidence:** 2

**Review:**

Overall, the author proposes a novel algorithm for learning Average-Reward MDPs. Compared with previous work, the algorithm achieves a near-optimal regret guarantee without prior knowledge of the diameter or the uniform mixing time.

However, I have the following concerns about this work:

1. The definition of the Average-Reward MDP is complex and unclear. For instance, there is no formal definition provided for key terms such as the diameter and hitting time.

2. Although the proposed algorithm achieves a near-optimal regret, it is important to note that the algorithm relies on a generative model that provides the transition dynamics for arbitrary state-action pairs. When comparing this with previous work in Tables 1 and 2, it is necessary to indicate whether an algorithm uses a generative model to ensure a fair comparison.

3. In the Proof Sketch section, the author discusses the Bernstein-type concentration analysis for the proposed algorithm, which is the key part in achieving near-optimal regret. However, another important contribution of this work is that it is parameter-free. It would be beneficial to include further discussion on how the algorithm removes the requirement for prior knowledge of the diameter while maintaining optimal performance.

4. The claim in the introduction that "finite sample properties of average-reward plug-in approaches have never been theoretically examined" seems too restrictive. As shown by the author in the table, several algorithms have already achieved finite sample complexity.

**Paper Award:**

No

---

> ### Author Response · Authors · 2024-11-23
>
> Thank you for your positive review. We would like to respond to your particular concerns:
> 1. The diameter is defined on page 5 in the average-reward criterion paragraph, although it is defined in terms of the hitting time $\eta_s = \inf \\{t \geq 1 : S_t = s\\}$ which as you mentioned was not included. We will add this definition in the final version.
> 2. As mentioned in the table caption, all prior work shown in Tables 1 and 2 also consider the same generative model setting as our paper. We will make this more clear.
> 3. Thank you for this suggestion; we agree that being parameter-free is an important contribution. At a high level, the plugin approach does not have any parameters which need to be set and thus naturally does not require any prior knowledge. To answer your question of why the optimal diameter-based rate is still possible, we first note that the only prior work which achieves the optimal diameter-based rate is [Zurek and Chen 2024]. We believe a main overall effect of using a discounted reduction with a relatively small and prior-knowledge-dependent effective horizon, as is done in [Zurek and Chen 2024], is to control variance parameters in a similar method to our Lemma 12. However, as shown in our Lemma 4, the variance parameters associated with the plugin approach (e.g. $\\|\widehat{h}^\star\\|_{span}$) are automatically controlled in terms of the diameter. In short, our analysis of the plugin approach shows it enjoys certain automatic adaptivity to the instance complexity, as opposed to requiring careful tuning of parameters to obtain the same adaptivity.
> 4. While prior works have introduced different algorithms with finite sample complexity bounds, none of them have studied the specific algorithm considered in our paper: the plug-in approach. In the intro of the paper, we argued that the plug-in approach is a particularly important algorithm to study.
>    - In more detail, the plugin approach is a generic template for constructing estimators for a functional of an unknown distribution/model (by plugging the empirical distribution/model into the functional) which is broadly used beyond RL and has been studied abstractly by statisticians in terms of its asymptotic/”infinite-sample” properties, hence our specific use of the phrase “finite-sample properties”. (It is not actually clear whether even standard asymptotic arguments would apply to our setting, since the plugin approach is commonly studied under the assumption that the functional is continuous, whereas the average-optimal policy may be discontinuous in the parameters of the transition kernel $P$.) As discussed in the paper’s introduction, we believe that developing a theoretical understanding of the plug-in approach is of particular interest since it is a natural and simple algorithm which does not utilize parameters which must be tuned using prior knowledge (contrasting the well-studied discounted reduction approach).

---

### Official Review · Reviewer_UQ2c · 2024-11-09

**Rating:** 6
**Confidence:** 3

**Review:**

This paper makes solid progress with the sample complexity of plugin approaches for average-reward MDPs (AMDPs).

# Pros

- This is the first paper giving optimal sample complexity proofs to plugin-type approaches for the policy optimization problem in AMDPs. The plugin approach is the most natural and simplest model-based algorithm for the policy optimization problem in MDPs. Moreover, when the setting is AMDPs, the plugin approach is more natural (and maybe more useful from an algorithmic perspective) than the methods based on the reduction to DMDPs. Hence, the sample complexity of the plugin approach is a fundamental problem. This work, though still a little way from fully solving this fundamental problem, made considerable progress on this basic problem.

- This is the first paper giving optimal algorithms (in the sense of optimal sample complexity) w.r.t. diameter $D$ and/or uniform mixing time $\tau_{unif}$ of the AMDP without the requirement of prior knowledge of these parameters.

- This is the first paper proving the simplest plugin approach without reward perturbation is sample optimal for the full range of sample sizes in DMDPs. This is also the paper extending the range of $\epsilon$ to the full range for the optimal sample complexities w.r.p. $D$ or $\tau_{unif}$ in DMDPs.

- This paper has solid proofs with new techniques (though I only get to skim them through), and it is well-written.


# Cons

- The bound (Theorem 1) for the simplest plugin approach has dependencies on some unclear terms ($\|\widehat h^*\|_{span}$ and $\widehat \gamma^*$) and has the extra requirement of the connectivity of the estimated MDP.

- We usually want the bound to only depend on the parameters of the MDP ($S, A$, and like $H = \|h^*\|_{span}, D, \tau_{unif}$) or the problem ($\epsilon, \delta$ and $\gamma$ in DMDP). It will be have been better if we can bound $\|\widehat h^*\|_{span}$ and $\widehat \gamma^*$ (which appear in the bound of Theorem 1) by these parameters. A lower bound showing a higher order dependence on the common parameters, if it is the truth, is also welcome.

- It would also be ideal to eliminate the requirement that the estimated MDP is weakly communicative (e.g. prove it holds with probability $1-\delta$); or to prove for any polynomial samples $n$ and tolerance probability $\delta$, it is inevitable to exist a non-weakly-communicative estimated MDP with probability $> \delta$.

- The results (Corollary 5, Corollary 7, Theorem 8) that provide algorithms with optimal sample complexities w.r.t. $D$ or $\tau_{unif}$ don't require prior knowledge of these MDP parameters, but only in either one of the parameters.

- When the sample size $n$ is given, the accuracy of the resulting policy can be bounded by $n$, and the algorithms really don't depend on $D$ or $\tau_{unif}$.

- But when the required accuracy $\epsilon$ is given, the algorithms will actually depend on $D$ or $\tau_{unif}$. The algorithms are the plugin approach with the anchoring technique (and with or without reward perturbation), and the anchoring technique needs a parameter $\eta$, which is set to be $\frac{1}{n}$ in the algorithms. To obtain the certain accuracy $\epsilon$, since the proved sample complexities are in terms of $D$ or $\tau_{unif}$, the algorithms will have to set $n$ to depend on these parameters, which will be further used for $\eta$ in the algorithms. This leads to that the algorithms actually require prior knowledge of these parameters, if I understand correctly. Can the authors clarify it a bit?

- The hardness result (Theorem 14) seems a bit incomplete to me. It cannot show a necessity of $\|\widehat h^*\|_{span}$ term in the complexity.

- The hardness just proves a lower bound of $\epsilon = \omega(\sqrt{\frac{H \log(Hn)}{n}})$, but in a precise sense, to show, as said by the author, the $\|\widehat h^*\|_{span}$ term can not be removed from the bound in Theorem 1, what we need to prove is $\epsilon = \omega(\sqrt{\frac{(H+1)\log^3(\frac{SAn}{\delta (1 - \widetilde\gamma^*)})}{n}})$, if I understand correctly. Can the authors comment a bit on this?

- Another reasonable way to show the necessity of this term might be to prove a lower bound with this term directly appearing in the bound. It would be good if the authors can add some discussions on the challenges of obtaining such a bound.

- This paper only proves that the plugin approach, with the anchoring technique and with reward perturbation, achieves the optimal sample complexities w.r.t. $D$ or $\tau_{unif}$. But two important problems remain:

  - Can the simplest plugin approach itself achieve the optimal sample complexity? If not, what is the complexity? How may the techniques and results in this paper help us understand the near-optimality of the vanilla plugin method for AMDPs?

  - Can we improve the sample complexities for the plugin approach with the anchoring technique and with reward perturbation, by replacing the $D$ or $\tau_{unif}$ in the bound with the parameter $H = \|h^*\|_{span}$? It's known that $H$ is smaller than both $D$ and $\tau_{unif}$.

# Some typos

- Sec 1.1 Line 8: should be $\widetilde O(\frac{SA \|\widehat h^*\|_{span}}{\epsilon^2})$ instead of $\widetilde O(\frac{SA \tau_{unif}}{\epsilon^2})$

- After Corollary 5 Line 4: should be Corollary 5 instead of Theorem 5

- Theorem 13 Line 2: should be Algorithm 2 instead of Algorithm 9

**Paper Award:**

No

---

> ### Author Response · Authors · 2024-11-23
>
> Thank you for your positive review. We also thank you for bringing these typos to our attention and we will promptly fix them. We now respond to some of the cons you identified:
>
> - While it is true that Corollary 5 achieves optimal complexity with respect to $D$ alone and Corollary 7 achieves optimal complexity with respect to $\tau_{unif}$ alone, Theorem 8 achieves the optimal complexity with respect to both parameters simultaneously (whichever is smaller), without needing to know either parameter. (This is because we can use the same steps as in Corollaries 5 and 7 to bound the $\min$ term in Theorem 8 by $\min \\{ D, \tau_{unif} \\} $. We will clarify this in the discussion after Theorem 8.)
> - Regarding the necessity of prior knowledge, we agree that the difference between $n$ and $\varepsilon$ is a subtle point with interesting implications. We have focused on the setting where a fixed-size dataset is provided (the fixed-$n$ setting). At a high level, many results in statistical learning are of a similar form, which provides an error guarantee conditional on the number of samples and values of the (unknown) complexity parameters. As we argue below, the fixed-$\varepsilon$ setting, where the algorithm must choose $n$ as a function of $\varepsilon$ *without* knowing the complexity parameter, generally appears more challenging. Our results are still the first to obtain such a fixed-$n$ guarantee with the optimal rate for the diameter and mixing-based settings without requiring prior knowledge. We also wish to provide some more detailed responses:
>   - Relationships between the $n$- and $\varepsilon$-based settings: We generally understand the $\varepsilon$-based setting to require more from an algorithmic standpoint. Our results for fixed $n$ intuitively answer the question “given a fixed-size dataset, how can I learn a good policy from it?”, and our results show that the plugin approach is able to do so optimally for the diameter and mixing-based settings. It seems like such a primitive would always be needed within an algorithm which is optimal for the $\varepsilon$-based setting, but in the $\varepsilon$-based setting the algorithm must also decide how much data to collect as a function of $\varepsilon$, that is how to choose $n(\varepsilon)$. As you point out, setting $n(\varepsilon)$ can be done using prior knowledge of complexity parameters. In order to avoid using prior knowledge, one approach to choosing $n(\varepsilon)$ is to try to estimate the complexity parameters. Another approach is to increase $n$ (e.g. geometrically), each time running the algorithm which is optimal for fixed $n$, and then performing some $\varepsilon$-optimality certification process. Thus to avoid prior knowledge use it seems we would need to combine the optimal fixed-$n$ algorithm with a certification procedure or a complexity parameter estimation procedure, and even if such procedures exist they may also require more samples than the chosen $n(\varepsilon)$ (that is, the certification/parameter estimation may require more data than the policy learning phase, causing the rate to be suboptimal).
>   - Concrete considerations for average-reward settings: The approach of estimating complexity parameters to remove prior knowledge requirements has been considered in some prior work. [Zurek and Chen 2024] and [Tuynman et al. 2024] both show that the span parameter $H =\\| h^\star\\|$ cannot be estimated within a constant factor using a sample complexity depending on $S,A,H$, preventing this approach from working with optimal $H$-based complexity. Even with perfect knowledge of $P$ it is generally intractable to compute $\tau_{unif}$, a mixing time bound for all policies (although in some special settings it may be possible to find useful upper bounds). Therefore, it also seems impossible to estimate $\tau_{unif}$. [Tuynman et al. 2024] use a diameter estimation procedure to estimate $D$ and achieve a bound for the $\varepsilon$-based setting, however their diameter estimation procedure builds a model estimate with nontrivial total variation distance from the true model and requires $D^2 S^2 A$ samples to do so, which may be much larger than the number of samples $DSA/\varepsilon^2$ required to simply learn an $\varepsilon$-accurate policy due to the higher-order dependence on $S$.

---

> > ### Author Response · Authors · 2024-11-23
> >
> > - We would like to clarify several points regarding the hard example for the plugin approach (Theorem 14).
> >   - Our theorem does in fact imply a result like $\varepsilon = \omega \left( \sqrt{\frac{(H+1)\log^3 (\frac{SAn}{\delta (1-\widehat{\gamma}^\star) } ) }{ n } } \right)$. In the hard example we construct, it is possible to calculate that $\frac{1}{1-\widehat{\gamma}^\star} = O(n^3)$, which we will add to the paper. Therefore, in the statement of Theorem 14, this $\frac{1}{1-\widehat{\gamma}^\star} = O(n^3)$ term can be absorbed within the other $n$ term in the log factor. The terms $S, A, \delta$ are all constant w.r.t. $n$ so they can also be absorbed. Also, as stated in the first part of the theorem, the hard example has $H=1$, so we can replace $H$ by $H+1$. For a dependence on $\log^3$ rather than $\log$, we note that actually the first part of the theorem states that (on the constant-probability bad event) the policy $\widehat{\pi}$ output by the plugin approach will have suboptimality $\Theta(1)$, which actually implies that any bound of the form $C \left( \frac{H \log(n H)}{n}\right)^{\alpha}$, for any exponent $\alpha > 0$ and any constant $C>0$, is false.
> >   - We also note that on the bad event, we have $\\|\widehat{h}^\star\\| = \Theta(n)$ (we will add this in the paper), and our theorem shows that this $\Theta(n)$ term is necessary.
> >   - It may be of interest to construct an example where $\\|\widehat{h}^\star\\|$ grows at a different rate other than $n$, but this seems more complicated due to the need for multiple parameters for the hard instance.
> >   - Overall, we thank you for your careful consideration of Theorem 14 and its implications, and we will try to incorporate the above comments to clarify these points.
> > - It may not be the case that the empirical MDP $\widehat{P}$ is weakly-communicating, and an example of the form you describe can be constructed, with details given below. We note that even in the case that $n$ is much smaller than the diameter and thus the empirical MDP is likely not (weakly-)communicating, our anchoring-based theorems (ex. Theorem 3) give nontrivial error guarantees in terms of span-based quantities. Dealing with the situation that the empirical MDP is non-weakly-communicating is thus one concrete advantage of using the anchoring technique, which is a simple way of enforcing weak-communicativity.
> >   - Details of construction: Let the true MDP $P$ be communicating and have two states, but they each are only connected to the other through a rare transition of probability $1/T$. Then this MDP will have diameter $T$, and we will need order $n=T$ samples to have a constant probability of observing both of these transitions at least one time in the dataset.

---

### Official Review · Reviewer_HSEG · 2024-11-11
**Good paper overall**

**Rating:** 7
**Confidence:** 3

**Review:**

The paper provides an analysis of the plugin method for solving average-reward MDPs with generative sample access. It shows that this canonical method achieves optimal sample complexity in the diameter and uniform mixing time setting. Importantly, these guarantees are achieved without the requirement to know problem-dependent quantities, a limitation of other methods based on reduction to discounted MDPs. The key ingredient in the analysis is a new error decomposition which recursively expands the error with higher-order terms. This technique also leads to sharper sample complexity bounds in the discounted MDP setting.

The paper is well written and overall easy to follow. The authors do a good job putting their results and techniques in the context of prior work. The paper also manages to convey good intuition for required techniques that enabled the new sample complexity results, in particular in the proof sketch in Section 4.

Optimal sample complexity results for the simple plugin method are a significant contribution. Many algorithmic and analytical techniques are similar to prior work, e.g. anchoring, reward perturbation or the higher-order error expansion. Still, details are different and those details matter. Certainly, showing that an anchoring probability of 1/n is sufficient is a very nice result. This paper offers a valuable set of tools that are likely to be useful in other settings and future research. All presented results are plausible, and no technical errors were identified in the proof sketch or a brief skim of the full proofs in the appendix. Overall, this is a strong paper that merits acceptance.

**Paper Award:**

No

---

> ### Author Response · Authors · 2024-11-23
>
> We thank you for your positive review. In response to your comment that the technical tools within our paper are likely to find use in other settings, we agree and would like to offer some more details. While, as you mention, “higher-order error expansions” are used in other papers and settings e.g. [Li et al. 2020], looking beyond the superficial similarities, our error expansion techniques are highly different to those of [Li et al. 2020]. In particular, our techniques do not have any implicit restrictions on the effective horizon, which is essential for the average reward setting (which can be viewed as an arbitrarily large effective horizon). We believe our techniques are also simpler and more robust, for example evidenced by the fact that our techniques are compatible with the simpler leave-one-out construction of [Agarwal et al. 2020] as well as the construction within [Li et al. 2020], whereas the error expansion arguments of [Li et al. 2020] are only compatible with the particular leave-one-out construction developed in their paper. We therefore hope that our techniques may enable future research developments by virtue of their greater flexibility and wider applicability (especially for average-reward settings).

---

### Meta-Review · Area_Chair_Git6 · 2024-12-14

**Recommendation:** Accept
**Confidence:** 5

**Metareview:**

This paper provides an analysis of the plugin method for solving average-reward MDPs with generative sample access. In particular, compared with previous work, the algorithm achieves a near-optimal regret guarantee without prior knowledge of the diameter or the uniform mixing time.

All reviewers like the paper. The AC agrees and recommends acceptance.

**Paper Award:**

No